# Study of the Extremely Low-Frequency Noise Characteristics of a Micro-Thrust Measurement Platform

**DOI:** 10.3390/mi15040508

**Published:** 2024-04-08

**Authors:** Liexiao Dong, Shixu Lu, Luxiang Xu, Ning Guo, Mingshan Wu, Shengtao Liang, Jianfei Long

**Affiliations:** 1School of Fundamental Physics and Mathematical Sciences, Key Laboratory of Gravitational Wave Precision Measurement of Zhejiang Province, Taiji Laboratory for Gravitational Wave Universe, Hangzhou Institute for Advanced Study, University of Chinese Academy of Sciences, Hangzhou 310024, China; dongliexiao22@mails.ucas.ac.cn (L.D.); lushixu21@mails.ucas.ac.cn (S.L.); guoning@ucas.ac.cn (N.G.); liangshengtao1994@163.com (S.L.); longjianfei@ucas.ac.cn (J.L.); 2University of Chinese Academy of Sciences, Beijing 100049, China; 3National Space Science Center, Chinese Academy of Sciences, Beijing 100190, China; 4Science and Technology on Vacuum Technology and Physics Laboratory, Lanzhou Institute of Physics, Lanzhou 730000, China

**Keywords:** micro-thrust measurement platform, vibration transmission, gravitational pendulum, numerical simulation

## Abstract

The critical structural parameters are optimized and studied using the numerical simulation method to improve the resolution and stability of the Micro-Thrust Measurement Platform (MTMP). Under two different ground random vibration environments, the parameters, such as pivot thickness, pendulum rod length, and pivot structure, are focused on analyzing the influence of the system’s resolution and stability. The results show that when the thickness of the pivot is 0.04 mm or 0.2 mm, and the pendulum rod length is 2 m, the effect of ground random vibration on the MTMP is minimized. At 0.1 mHz, it can reach 0.0057 μN/$\sqrt{\mathrm{Hz}}$. In the series double-pivot structure, an appropriate increase in the distance between the sheets can further optimize the above conclusions. The results and analysis within this study can provide support for the engineering design of the MTMP.

## 1. Introduction

Space exploration plays a key role as one of the essential means of human exploration of outer space [1]. As technology advances and scientific knowledge expands, more and more space science missions are being planned and added to the agenda. Space gravitational wave detection, as a primary space science mission, is of great significance to further explore the mysteries of the universe [2,3,4,5]. Currently, there are three ongoing missions to detect gravitational waves in space that have undergone phase I validation, namely, the LISA program conducted by the European Space Agency (ESA) in cooperation with the National Aeronautics and Space Administration (NASA), the “Taiji” Program led by the Chinese Academy of Sciences (CAS), and the “Tianqin” Program led by a consortium of universities, with Sun Yat-sen University as the main body [6,7,8]. At the same time, the precision of satellite attitude control and position holding is the key to all gravitational wave detection missions in space [9]. Drag-free control technology and satellite thrusters can provide an exact and stable platform for laser interferometry to obtain gravitational wave information [10]. As the primary actuator of drag-free control, the performance of the micro-thruster directly determines the performance of the drag-free control platform, so different gravitational wave detection missions have specific requirements for the thrust resolution and range of the micro-thruster. In addition to the optimized design of the thrusters themselves, the requirements for their measurement platforms are also one of the keys for different types of aerospace thrusters [11].

A series of micro-thrust measurement devices have been proposed internationally to measure the micro-thrusters, and the main structures include inverted pendulums, gravitational pendulums, torsion pendulums, composite pendulums, and so on [12]. Denis Packan and Jean Bonnet of The Office National d’Etudes et de Recherches Aérospatiales (ONERA) have developed an inverted pendulum-based micro-thrust measurement device with a thrust measurement range of 1 μN to 700 μN and a noise power spectral density of less than 0.1 μN/$\sqrt{\mathrm{Hz}}$ between 0.01 Hz and 1 Hz [13]. The Laser Interferometer Space Antenna (LISA) team designed a pendulum structure micro-thrust measurement device with a thrust test range of up to 1 mN, a resolution that can reach 0.1 mN, and a background noise in 0.01 Hz to 1 Hz that can be less than 0.1 μN/$\sqrt{\mathrm{Hz}}$, and it was tested employing the cold gas thruster and field emission electric thruster [14]. Hao Xu of the Huazhong University of Science and Technology studied a micro-thrust measurement device belonging to the composite pendulum with a measurement range of 1000 mN, and a noise of less than 0.1 μN/$\sqrt{\mathrm{Hz}}$ in the range of 0.01 Hz to 1 Hz [15]. Chao Yang of the Institute of Mechanics, CAS, developed a torsion pendulum-type micro-thrust measurement device with a thrust range of 0 μN to 400 μN and a resolution of 0.1 mN, and the background noise power spectral density at 0.01 Hz to 1 Hz is better than 0.1 μN/$\sqrt{\mathrm{Hz}}$. Yuanxia Yang of Huazhong University of Science and Technology achieved a thrust range of up to 264 μN with a resolution of 0.09 μN using a unique suspension design and precision assembly of a torsionally balanced structure [16].

The resolution of the micro-thrust measurement device mentioned above can reach the order of micro-Newtons or even sub-micro-Newtons, and the background noise is also at an excellent level under the specific thrust test environment. However, many significant challenges remain. For example, the raw data obtained from testing do not directly indicate the resolution level, and require further processing by additional methods. Furthermore, the portability of these data processing methods across different test environments is limited. Existing studies are mostly limited to the design optimization of the structure, but a few studies have been reported on the test environment around the micro-thrust measurement device. In general, to further optimize the micro-thrust measurement device to satisfy the thrust noise requirement of the “Taiji” program of better than 0.1 μN/$\sqrt{\mathrm{Hz}}$ in the frequency band of 0.1 mHz to 1 mHz [17], the relationship between the thrust measurement device and its surrounding test environment is the key to be investigated.

Generally speaking, the temperature, airflow, and ground vibration in the external environment are the three main factors affecting micro-thrust measurement. Among them, since the micro-thrust measurement device and the micro-thrusters work in a high-vacuum environment, and placing heat insulation materials to separate the device and the environmental platform can meet the requirements of the test environment temperature, building a simple test environment can minimize the impact of temperature and airflow. However, in contrast, the vibration of the whole thrust test platform caused by ground vibration is unavoidable.

There have been many findings in related research areas on vibration transfer using numerical simulation. Manzato et al. investigated a control scheme for a three-bladed control advanced research turbine. They reported preliminary results of an extensive full-scale modal test, which will be used to develop an advanced controller to optimize power and minimize structural loads [18]. Zaghbani et al. demonstrated how OMA can be utilized in industry. The method was successfully applied during the high-speed machining of 7075-T6 aluminum alloy [19]. Pavlović et al. performed a modal analysis by varying the structural design of a large-scale machine tool for polishing and lapping tiles and ultimately optimizing the intrinsic frequencies of the different design alternatives [20]. A. Martini et al. built a new machine tool structural finite element model. They experimentally verified it on a prototype machine, and the simulation results are consistent with the data provided by the experimental modal analysis, further proving the simulation’s feasibility [21].

Based on the principle of gravitational pendulum, this paper designs a Micro-Thrust Measurement Platform (MTMP) for micro-thrusters. Because of the high efficiency and economy of the numerical simulation method, numerical simulation research is carried out for its critical structure. In order to reduce the influence of random ground vibration on the resolution of thrust measurement and the long-term stability of MTMP, the influence of significant parameters such as pivot thickness, pendulum rod length, and pivot structure on the thrust measurement is investigated based on the numerical simulation software COMSOL, and the corresponding results are analyzed and interpreted.

The rest of the paper is organized as follows: Section 2 provides the base model, theory, and boundary setup. Section 3 presents the simulation results and draws preliminary conclusions. Section 4 analyzes and discusses the simulated results in more detail. Finally, Section 5 concludes the work.

## 2. Models and Methods

### 2.1. Pendulum Structure

The pendulum frame MTMP consists of a 120×120×2400mm “T” aluminum frame fixed to an optical platform below as a support rod. The carbon fiber pendulum rod is rigidly connected to the beryllium bronze sheet through the lower connector, and the beryllium bronze sheet is rigidly connected to the support rod through the upper connector, which ensures that the pendulum rod operates in a single direction, with the direction of rotation parallel to the plotting plane in Figure 1. The symmetrical placement of the two carbon fiber rods about the support rod is designed to ensure the symmetry of the overall MTMP structure and to facilitate the differential processing of the actual thrust data. The measurement point of the MTMP is located at the end of the bottom loading platform, which is used to monitor the displacement change before and after the thrust is applied. In addition, a wireless accelerometer is mounted on top of the support rod to detect its attitude and adjust it by adjusting the optical platform. It is easy to see that the components of the MTMP are screwed together to form a rigid connection. Ideally, when a small external force is applied, only the pivot will deform significantly, and the pendulum rod will only oscillate in a direction parallel to the plotting plane.

Beryllium bronze sheet is a typical pendulum pivot component with high tensile strength and fatigue resistance. Moreover, carbon fiber has excellent properties such as high strength, low mass, high Young’s modulus, and very low coefficient of thermal expansion, which significantly reduce the additional error caused by temperature changes [22]. Figure 1 shows the general structure and some details of MTMP. Table 1 shows the basic structural parameters of the MTMP.

### 2.2. Principle of Micro-Thrust Measurement

The MTMP core structure discussed in this paper is similar to two symmetrically arranged single-degree-of-freedom gravity pendulums. Figure 2 shows the schematic diagram of the gravity pendulum, which constitutes a typical second-order system whose mathematical equations can be described as follows [15]:(1)$$I\ddot{\theta }+\lambda \dot{\theta }+K\theta =\tau$$
where *I* and θ represent the rotational inertia and angular change value of the pendulum rod, respectively; *K* represents the angular stiffness of the pendulum; λ is the damping factor of the system; and τ is the applied torque. When the thrust force is applied to the pendulum, the external thrust force can be introduced into the above-mentioned mathematical model, which is obtained [15]:(2)$$I\ddot{\theta }+\lambda \dot{\theta }+K\theta =\tau =F_{0}\cos\left(\omega t\right)L$$
where $F_{0}$, *L*, and *w* represent the value of the applied thrust force, the length of the thrust force arm, and the frequency, respectively. Ideally, the pendulum rod produces a tangential acceleration due to the combined effect of the tangential component of its own gravity Mgsinθ, and the thrust force $F_{0}\cos\left(\omega t\right)$. With a small amount of angular change, the tangential component can be simplified to Mgθ. However, for a constant force applied to the pendulum, ω equal to 0, the pendulum rod will reach an equilibrium state at the new position, yielding a second mathematical equation:(3)$$Mg\theta H=F_{0}L$$
where *H* represents the distance from the pendulum rod’s center of mass to the suspension point, which is the length of the force arm of gravity. The model is solved by combining the above mathematical equations, which can be obtained by solving:(4)$$F_{0}=K\delta \theta =K_{f}\delta u$$
where δθ and δu represent the change of angle and displacement after the pendulum rod arrives at the equilibrium position under the action of thrust, and $K_{f}$ represents the displacement expressions of stiffness. It is worth noting that in the case of a small amount of angular change, *K* and $K_{f}$ are approximately equal.

It is easy to see that the above model is built on the premise that the pivot’s stiffness has little effect on the pendulum frame as a whole, so the pivot structure is analyzed and discussed next. Since the pivot sheet connecting the pendulum rod and the support rod is a rectangular cross-section body configuration, the stiffness of the sheet can be calculated with the Navier structure theory, and the stiffness expression is [23]:(5)$$K_{s}\approx EC^{3}D/12L_{s}$$
where *E* is the Young’s modulus of the material, and *C*, *D*, and $L_{s}$ represent the thickness, width, and length of the sheets. The estimation of all the used sheets reveals that the sheets used in this paper are classified into two cases: one is that their own stiffness is much less than the stiffness generated by the weight of the pendulum rod itself, the weight of the pendulum rod is used as the source of stiffness, and the above model is valid. The other case is that the stiffness of the sheet is not negligible, which affects the overall stiffness of the system, and it is necessary to superimpose the gravity and the stiffness of the sheet itself in the above model to obtain the final system stiffness. It is worth noting that the stiffness of the pendulum mentioned in this paper is adjustable in size, and the stiffness of the pendulum can be adjusted by adjusting the counterweight and the position of the center of mass of the pendulum rod.

### 2.3. Vibration Transfer Analysis

Vibration may affect a measurement system or environment in different ways, and the theory of vibration transfer is designed to study structural mechanics. Generally, it is possible to study how vibration is transmitted and affected within a structure by system transfer functions, modal analysis, and finite element analysis [24]. The eigenvalues of a system are critical in vibration transfer problems. Through the eigenvalues, the system’s eigenfrequencies and modes of vibration in different degrees of freedom are obtained. Furthermore, the system’s modal shape and frequency response can be obtained, which is crucial for determining the system’s response under a particular excitation. For systems without additional damping, the eigenvalue problem can usually be formulated and solved by the following mathematical Equation [25]:(6)$$[K-\omega^{2}M]u=0$$
where K is the stiffness matrix, M is the mass matrix, u is the eigenmode displacement vector, and ω=2πf is the angular frequency.

Modal analysis is as important as eigenvalues. Modal analysis is a common method used to study the dynamic behavior of structures. It analyzes the vibration characteristics of a system by decomposing the structural vibration modes. The basic equations in the modal analysis are [25]:(7)$$M\Phi \ddot{q}+K\Phi q=0$$
where Φ is the mode shape matrix, which contains information about the vibration shapes of the vibration modes, and *q* is the modal displacement vector, which describes the vibration of each mode. The motion of a vibrating system is described by transforming the dynamical equations of the system into modal space. The goal of modal analysis is to solve this equation to obtain the mode shape matrix Φ and the corresponding mode frequencies, as well as information on each mode’s mode shape and amplitude.

### 2.4. Research Objects

Under the test environment of high vacuum and heat transfer isolation, the amount of change in the pendulum rod position is only related to the vibration transmission ability of the pendulum frame. Therefore, this paper optimizes the pendulum frame’s vibration isolation ability by changing its structure’s critical parameters to reduce the negative impact on the resolving power and stability of the pendulum frame due to ground vibration.

Figure 3 shows the vibration transfer roadmap. It is easy to see that each part of the MTMP plays a key role. The stiffness of each part affects its own and the overall intrinsic frequency. Structures with higher stiffness typically have higher intrinsic vibration frequencies, resulting in vibration energy propagating locally or being transmitted over a smaller area. On the other hand, structures with lower stiffness have lower intrinsic vibration frequencies, which may result in a broader spread of energy. At the same time, different intrinsic frequencies have different sensitivities to different frequency bands of noise in random vibration signals, and the resulting resonance phenomenon is also a factor that cannot be ignored.

Therefore, we can optimize the overall vibration isolation capability of MTMP by changing the material and size of the support rod, the thickness of the upper and lower connectors, the thickness of the pivot, the structure of the pivot, the length of the pendulum, and material of the pendulum. Further analysis shows that the upper and lower connectors only play the role of a connection and are rigidly connected with other parts. The change in their thickness has the most minor effect on the vibration isolation capability of MTMP. The pivot, as the most sensitive structure in the overall structure of the MTMP, has the most significant influence on the thickness and distribution of the pendulum, and the length of the pendulum rod determines the stiffness of the whole measurement system, which is one of the critical factors affecting the transmission of vibration. Therefore, under the condition that the overall structure and material remain unchanged, this paper focuses on the thickness of the pivot, the structure, the pendulum rod’s length, and other vital parameters affecting the MTMP.

### 2.5. Boundary Settings

The simulation models in this paper are solved using the random vibration study in the COMSOL 3D Solid Mechanics module. Random vibrations usually do not have a definite periodic pattern. Irregular and random excitations, such as those caused by ground vein vibrations or nearby artificial activities, are responsible for these effects. Therefore, they are usually characterized by analyzing the frequency spectrum and Power Spectral Density (PSD) of the vibration signal to judge the vibration transmission capability of different structures. After inputting a random vibration signal at the source, the random vibration will propagate through the entire structure.

First, the random vibration can be decomposed into the superposition of multiple thrust signals, corresponding to multiple random loads acting on the structure, which also have different degrees of interdependence among them. Meanwhile, through the analysis in Section 2.1, the pendulum frame structure can be regarded as a single-degree-of-freedom vibration model, and the vibration generated by the pendulum frame structure will be consistent with the mathematical model of the spring-mass-damping system, whose mathematical model is the form given in Equation (Equation 2).

In general, the results of the random vibration analysis of structures can only be interpreted statistically, such as root mean square (RMS), PSD, etc. Random vibration analysis in COMSOL involves the calculation of the eigenmodes of the overall structure of the pendulum frame, as well as simplifying the model based on the superposition of the modes and the reduced-order modeling (ROM) function in solid mechanics. In addition, the core of determining the magnitude of the system’s vibration transfer capability lies in evaluating the simulation analyses’ results.

Before performing the vibration transfer capacity calculations for the overall structure, the material and boundary settings in the COMSOL Solid Mechanics module are performed. Firstly, the materials of each part of the components of the pendulum frame are selected, and Young’s modulus, density, and Poisson’s ratio of each material are supplemented separately. Secondly, the four supporting surfaces of the optical platform are set as fixed constraints, and boundary loads are set at the bottom of the support rods for simulating external vibration inputs. The mesh uses a free tetrahedral mesh with hyperfine division for general solution regions, such as the pendulum rod, to improve the computational speed and excellent division operations for key solution regions, such as the pivot and pendulum rod.

### 2.6. Criteria for Judging

In this paper, the PSD and RMS of the displacement at the test point are used as the output parameters. The RMS is used to characterize the random displacements and their probability distributions generated under the current vibration environment and then to evaluate their effects on the resolution of the pendulum frame. The PSD is used to characterize the response of the pendulum frame system in different frequency domains, which in turn is used to evaluate the background noise of the MTMP.

## 3. Results and Analysis

### 3.1. Real Ground Vibration Input

Ground vibration is a continuous phenomenon caused by various natural and anthropogenic factors. Natural factors include the movement of crustal plates, such as ground vibrations caused by earthquakes, storms, lightning strikes, or volcanic activity [26]. At the same time, anthropogenic factors such as transport, construction activities, and industrial production are also important sources of ground vibration. In order to ensure the authenticity of the simulation results and the guidance significance to the actual structure, two real ground vibration situations under different environments are selected as the vibration inputs to the whole simulation model. The conclusions obtained from different vibration environments are verified with each other. Both signals are ground detected and acquired by Guralp’s seismometer. To harmonize with the units of evaluation of the pendulum frame stability index, the acceleration signal acquired by the seismograph is converted to the Newman force form, which is used as the input for random vibration. Newton’s second law converts the acceleration signal acquired by the seismograph to a force excitation signal:(8)$$P_{F}\left(f\right)=m_{t}\;\cdot \;P_{a}\left(f\right)$$
where $P_{F}\left(f\right)$ and $P_{a}\left(f\right)$ is the PSD of the force excitation signal and acceleration signal.

As shown in Figure 4, the low background vibration signal (Low background vibration) is taken from the underground laboratory of Jinping, China, where the anthropogenic noise is low, and the high background vibration signal (high background vibration) is taken from the laboratory of Hangzhou Institute of Advanced Studies, University of Chinese Academy of Sciencesy (HIAS, UCAS). It is easy to see that there is an apparent difference between the ground signals reflected by these two signals. Also, due to the restrictions imposed on the MTMP structure in this paper, the pendulum rod has only one degree of freedom in one direction. Therefore, the input to the random vibration is also unidirectional. Meanwhile, during the test analysis, since the MTMP overall structure is relatively simple and does not have any dynamic loads, the resonance phenomenon caused by multiple excitations does not occur.

### 3.2. Modal Analysis

From the introduction in Section 2.1, it is easy to see that the index for judging the structural stability of the MTMP is limited to the displacement PSD at the measurement point. Also, since there is no dynamic load on the pendulum for the analysis of pendulum stability, it does not lead to the occurrence of resonance phenomena due to having more than one excitation. The model of the MTMP used for the simulation analysis is sufficiently simplified to estimate the mode shapes and eigenfrequencies correctly. The focus of this study is to investigate the effect of design variations on the stability of the MTMP. As shown in Figure 5, the vibration modes obtained for the pendulum frame with different combinations of structural parameters are essentially the same, and only the eigenfrequencies are changed.

It is not difficult to see that of the eight variations occurring in the structure, only four will have a significant impact on the results in the *Z*-axis direction of interest in this paper, but at the same time, due to the high strength and high modulus of elasticity of the carbon fiber rods, the resulting structural variations of the latter two bring much less impact than the first two, so the first two modal vibration modes have the most significant impact on the stability of the pendulum frame.

### 3.3. Pendulum Rod Length Simulation Results

Due to the requirements of the gravitational wave detection project, it is known that the maximum value of the thrust to be measured should reach 100 μN, and the resolution should reach 0.1 μN [27]. Therefore, the accuracy of the system puts forward certain requirements. When the pendulum rod is 2 m long, the stiffness generated by the pendulum rod’s own gravity is about 4 N/m, so the extra stiffness is not greater than 0.004 N/m when the overall measurement error of the thrust is less than 0.1 μN; that is, the stiffness of the sheet can be ignored. Meanwhile, as shown in Figure 6, a stiffness of 0.004 N/m corresponds to a 0.1 mm sheet. Therefore, centering on 0.1 mm, the used sheets are divided into two separate groups of contents to be studied separately.

The length of the pendulum rod not only determines its mass but also, due to the presence of the lower connector, affects the distance between the end measuring point and the center of mass of the pendulum rod. This distance is a critical factor in determining the stiffness of the pendulum frame test. In this section, the thickness of 0.04 mm is selected as the pivot among the sheets that do not affect the overall stiffness, and a total of five pendulum rod lengths of 2.3 m, 2 m, 1.7 m, 1.4 m and 1.1 m are selected to study their effects on the overall vibration transfer capability of the pendulum frame.

Figure 7a,b show the PSD of the pendulum measuring points in two random vibration environments from 0.1 mHz to 5 Hz. It is easy to see that the PSD of the pendulum measuring point in the low ground vibration environment is one order of magnitude better than that in the high ground vibration environment. As shown in Figure 7c, with the increase in the pendulum length, the displacement PSD value of the pendulum measuring point at 0.1 mHz to 1 mHz has an obvious tendency to decrease and then increase, and reaches the lowest value when the pendulum rod length is 2 m. This pattern has good consistency in two different random vibration background environments. By calculation, the optimal PSD of the displacement measuring point can reach 0.008 μN/$\sqrt{\mathrm{Hz}}$ at 0.1 mHz in the low-vibration environment and 0.12 μN/$\sqrt{\mathrm{Hz}}$ in the high-vibration environment.

As shown in Figure 7d, with the increase in the pendulum rod length, the RMS values of the displacement test points, although showing different trends in different random vibration environments, all reach the best effect at a pendulum stiffness of 2 m. Different vibration environments have different effects on the resolution of the pendulum frame. Specifically, in a low-ground vibration environment, the RMS value of the displacement of the pendulum measuring point is significantly lower than that in a high-ground vibration environment. However, it is worth noting that in both ground vibration environments, the random vibration signals transmitted to the displacement measuring points do not have more than sub-micron effects on the pendulum rod swing. By calculation, the minimum interference value of background vibration noise in the displacement spectrum time-domain signal is 0.008 μN in the low-vibration environment and 0.044 μN in the high-vibration environment.

Next, the thickness of 0.2 mm is selected as the pivot among the sheets that have some influence on the overall stiffness. Moreover, the same selected five sets of pendulum rod lengths of 2.3 m, 2 m, 1.7 m, 1.4 m and 1.1 m and their influence on the overall vibration transmission ability of the pendulum frame are investigated.

Figure 8a,b show the PSD plots of the pendulum measuring point under two random vibration environments from 0.1 mHz to 5 Hz. It can be found that with the increasing length of the pendulum rod, the PSD at the pendulum measuring point under two different random vibration environments shows an obvious trend of decreasing and then increasing, and an optimal solution exists when the pendulum rod length is 2 m. The PSD at the pendulum measuring point under two different random vibration environments shows a clear decreasing and then increasing trend. As shown in Figure 8c, 0.0057 μN/$\sqrt{\mathrm{Hz}}$ and 0.09 μN/$\sqrt{\mathrm{Hz}}$ can be achieved at 0.1 mHz under two different random vibration environments.

As shown in Figure 8d, with the increase in the pendulum rod length, the RMS values at the pendulum measuring points under different random vibration environments all have the obvious trend of first increasing, then decreasing, and then increasing. At the same time, the RMS value when the length of the pendulum rod is 2 m is extremely small, and there is not much difference between it and the minimum value in the test. By calculation, the interference value of random vibration in a low-vibration environment is 0.006 μN, and the interference value in a high-vibration environment is 0.01 μN.

### 3.4. Pivot Thickness Simulation Results

According to the above Navi structural stiffness formula calculations, it can be observed that the thicker the sheet, the greater its stiffness. However, when the sheet’s stiffness reaches a certain threshold, its impact on the overall stiffness of the pendulum frame becomes significant and cannot be ignored.

In this section, based on a pendulum rod length of 2 m, six sheets with thicknesses of 0.01 mm, 0.02 mm, 0.04 mm, 0.06 mm, 0.08 mm and 0.1 mm are selected as the pivot, and their influence on the overall vibration transmission of the pendulum frame is investigated.

Figure 9a,b depict the displacement PSD plots of the displacement test point under two different random vibration environments, ranging from 0.1mHz to 5 Hz. The background PSD of the displacement test point in the low random vibration environment surpasses the simulation output in the high random vibration environment. As the pivot thickness increases, the random vibration PSD of the displacement measurement point at 0.1 mHz to 1 mHz tends to decrease and then increase, reaching its minimum value when the pivot thickness is 0.04 mm. This observation is well demonstrated in Figure 9c, and these trends align consistently under two different ground vibration environments. The optimal value derived from this simulation section corresponds to the previous findings.

Figure 9d illustrates the effect of vibration on the resolution of the pendulum frame under two different random vibration environments. As the thickness of the pivot increases, the RMS values of the pendulum measuring points exhibit a fluctuating trend of decreasing and then increasing. However, the trend line fitting the change of RMS values only exhibits a local minimum value at 0.04 mm, showing minimal deviation from the global minimum value point located at 0.1 mm. Thus, the ground vibration’s impact on the resolution of the pendulum frame is minimized when the sheet thickness is 0.04 mm, validating the optimal value identified in the preceding section.

For the second group, also based on a pendulum rod length of 2 m, six sheets with thicknesses of 0.1 mm, 0.2 mm, 0.4 mm, 0.6 mm, 0.8 mm and 1 mm are selected as the pivots to study the effect of pivot thickness on the overall vibration transmission of the pendulum frame.

Figure 10a–c illustrate that with increasing pivot thickness, the PSD at the measuring points of the pendulum frame exhibits a clear trend of decreasing and then increasing under two different random vibration environments, and an optimal solution is observed at a sheet thickness of 0.2 mm. As depicted in Figure 10c, under two different random vibration environments, the PSD at 0.1 mHz reaches 0.0057 μN/$\sqrt{\mathrm{Hz}}$ and 0.09 μN/$\sqrt{\mathrm{Hz}}$ when the sheet thickness is 0.2 mm. Meanwhile, Figure 10d demonstrates that the RMS values at the pendulum measuring point are consistent at 0.2 mm and 0.6 mm under different vibration environments, with no significant differences observed. The interference from random vibration in the low-vibration environment measures at 0.02 μN, while the interference value in the high-vibration environment is 0.04 μN.

### 3.5. Pivot Distribution Simulation Results

In addition to the aforementioned studies, this paper conducts further investigations into the novel pivot structure to examine the impact of various configurations of the double-pivot structure on the overall vibration isolation performance of the system. The beryllium bronze sheets are inserted into the reserved spaces of the upper and lower connectors and are screw-locked to form the pivot structure. The upper and lower connectors have pre-drilled holes and are rigidly connected to the support bar and pendulum by screws. Adhering to the optimal research outcomes discussed earlier, this section primarily introduces two new pivot structures: left–right series and front–back parallel pivots. The specific structure is shown in Figure 11. We aim to assess the effects of three different pivot structures—single sheet, parallel double sheets, and series double sheets—on the overall vibration transmission of the pendulum frame.

As depicted in Figure 12, under two distinct random vibration environments, the displacement PSD values at 1 mHz and 0.1 mHz for the series-connected dual-pivot structure outperform those of the single-pivot and parallel-connected dual-pivot structures. Notably, the disparity in effect between the pivot structures is more pronounced in the background noise of random vibration on high ground. Furthermore, the single-pivot structure significantly outperforms the pivot structure with front and rear parallel connections.

For the aforementioned double-pivot structures, this paper delves deeper into the configurations by varying the distance between the two sheets in the double-pivot structure and the width and thickness of the sheet in the single-pivot structure. As depicted in Figure 13c,d, under two distinct vibration environments, the parallel double-pivot structures with sheet distances of 4mm and 10mm exhibit similar vibration transmission abilities in the pendulum frame. Moreover, the two double-pivot structures of 0.04mm demonstrate an equivalent effect to the single-pivot structure of 0.08mm as seen in Figure 13a,b.

The comparison between the single-pivot structure with a width of 2cm and the double-pivot structure with sheet widths of 1cm and spacings of 4mm and 14mm, respectively, is illustrated and analyzed. Notably, in two different random vibration environments, the double-pivot structure with varying spacings outperforms the single-pivot structure. Specifically, the double-pivot structure achieves optimal performance when the spacing is set to 4mm.

## 4. Discussion

### 4.1. Influence of Pendulum Frame Structures on Vibration Transmission

According to the description in the above section, it can be found that either with the change of the length of the pendulum rod or the thickness of the pivot, there is an obvious inflection point of the vibration transfer capability of the overall structure, i.e., the optimal solution of the system to shield the ground vibration. The merit of the system’s vibration transfer capability is affected by two key factors: damping ratio and eigenfrequency. The damping ratio in the vibration system can directly affect the degree of attenuation of the vibration transmission in the system; the specific relationship between the formula is
(9)$$\zeta =\frac{c}{2\sqrt{mk}}$$

From the vibration transfer diagram in Figure 3, it can be seen that the damping ratio between each part of the pendulum frame belongs to the series relationship, and the change of the damping ratio of each part of the structure in the middle has a key influence on the vibration transfer effect of the whole system. In this paper, by changing the length of the pendulum rod and the thickness of the pivot, the mass and stiffness of the pivot and the pendulum rod in the system are changed, which in turn affects the overall damping ratio of the system.

The eigenfrequency of each part of the pendulum frame is also an important factor affecting the vibration transmission. The eigenfrequency solution for a single-degree-of-freedom vibration system can be estimated by the following equation:(10)$$f_{e}=\frac{1}{2\pi }\sqrt{\frac{k}{m}}$$
where $f_{e}$ denotes the eigenfrequency. Resonance occurs when the external excitation frequency closely matches the eigenfrequency of the system. In a state of resonance, the vibration amplitude of the system can significantly increase even with a small external excitation force.

Pendulum frames constructed with pivots of varying thicknesses and pendulum rods of different lengths exhibit distinct eigenfrequencies. Consequently, they display different sensitivities to vibration signals across various frequency bands of random vibration. This sensitivity directly influences the overall vibration transfer capability of the pendulum frame.

Meanwhile, the actual damping of carbon fiber rods as a composite material is challenging to estimate. Marco Troncossi et al. have quantitatively evaluated the effect of viscoelastic materials on the modal parameters of carbon/epoxy sheets through experimental modal analysis and derived an approximate damping model based on measured data [28].

In this paper, we refer to Qin Tengfei et al.’s conclusions on the shear strength and damping properties of short beams of carbon fiber composites fabricated by the composite molding methods of hot compression sterilization and hot patching to set up the damping model of carbon fiber rods in the article [29].

The inflection point of the system vibration transmission capacity obtained by numerical simulation is the optimal solution of the two factors of the pendulum system structure damping ratio and eigenfrequency in different vibration environments. It is easy to see that, among the ten sets of results, 2m and 0.04mm are the optimal solutions for the length of the pendulum rod and the thickness of a single pivot, respectively, and the above conclusions are the same in different vibration environments.

Due to the complexity of the real pendulum frame structure, it is difficult to solve the quantitative damping ratio and eigenfrequency for each part, so this paper carries out a design study on the pivot and the pendulum rod, which are the two key structures of the pendulum frame. By solving different parameter combinations to obtain the characteristic value of resolution (RMS) and the characteristic value of stability (PSD), the optimal combination of the structure can be obtained after comparison. Under the premise that the ground vibration has an acceptable impact on the thrust resolution, the most stable pivot and pendulum rod combination system is finally obtained to offset the impact of the ground vibration on the background noise of the pendulum frame itself and the thrust measurement as much as possible. In the subsequent work, the focus of future work is to quantitatively assess the vibration shielding ability of the rest of the real pendulum frame structure and form a set of systematic vibration transfer theories.

### 4.2. Effect of Pivot Structure Distribution on Vibration Transmission

Firstly, the vibration shielding capabilities of different double-pivot structures exhibit substantial differences. Regarding the left–right series pivot structure, its double-pivot configuration contributes to an overall increase in the pivot structure stiffness and mass. This alteration impacts the damping ratio and eigenfrequency of this structure segment, consequently altering the system’s overall vibration isolation level. Notably, this structure modification is akin to adjusting the sheet width to manipulate the overall stiffness. Compared to the sheet length and thickness, the width exerts the least impact on stiffness, allowing for finer adjustments. This aligns with the ability to perform more nuanced adjustments in stiffness, analogous to fine-tuning the structure’s eigenfrequency.

Furthermore, symmetrically positioning the two sheets along the pendulum rod’s front–back centerline enlarges the effective pivot structure width. This elongation enhances the contact length between the pivot structure itself and the adjacent pendulum frame structures above and below, mitigating the single pivot structure’s tendency to overturn due to ground vibration. Consequently, this configuration bestows greater stability and improves shielding against random ground vibration noise. However, an optimal distance between the two sheets exists, where excessively large or small distances degrade the pendulum frame’s ability to shield against ground random vibrations. This phenomenon is well substantiated by simulation results, demonstrating a correlation between increased lamina distance and this impact.

Two disparities exist between the double-pivot structure with a front–back parallel connection and the left–right series-connected double-pivot structure. Firstly, the front–back parallel structure’s resemblance to altering sheet thickness greatly influences the overall structure’s vibration transmission capability, with thickness changes exerting a more pronounced effect than width adjustments. This reaffirms the conclusion that vibration isolation effectiveness diminishes as the sheet thickness exceeds 0.04 m. Secondly, the pivot structure is structurally enhanced by a front–back parallel pivot arrangement, symmetrically positioned around the pendulum rod’s left–right centerline. This configuration effectively widens the plane of the pivot structure to other structures. However, resembling a parallelogram frame, its stability is notably inferior to that of the single-pivot structure and the left–right series-connected double-pivot structure. Consequently, this configuration introduces additional pivot thickness and an unstable pivot structure without effectively shielding against pivot-induced flipping in the opposite direction, akin to the impact of increased single-pivot structure thickness. Adjusting the relative distance between the two sheets fails to optimize the shielding capability further due to inherent structural limitations as corroborated by the simulation results.

## 5. Conclusions

This paper reveals the vibration transfer characteristics of the MTMP with its core unit under different structural parameters. The optimal combination of parameters is found among 24 sets of different combinations and verified by two accurate ground vibration signals, respectively. In addition, a new pivot structure is proposed to improve the stability of the device without introducing additional errors. When the pivot thickness reaches 0.04 mm or 0.2 mm, and the pendulum rod length is 2 m, the overall vibration isolation effect of MTMP is optimal. At 0.1 mHz, it can reach 0.0057 μN/$\sqrt{\mathrm{Hz}}$. The proposed series pivot structure can further enhance the vibration isolation capability of the system, reaching 0.001 μN/$\sqrt{\mathrm{Hz}}$.

In summary, without considering human factors such as installation errors and other external environmental disturbances such as temperature, the random ground vibrations in the underground laboratory in Jinping, China, can meet the target range of the sub-micro-Newtonian thrust measurement task, both in terms of the long-term stability of the pendulum frame and the impact on the measurement resolution. However, for the Hangzhou Institute of Advanced Studies of UCAS, the ground vibration level particularly impacts the device itself. Based on the above conclusions, more in-depth research will be carried out on other important factors affecting the pendulum rod test, such as the ground vibration isolation device and the ambient temperature.

## Figures and Tables

**Figure 1 micromachines-15-00508-f001:**
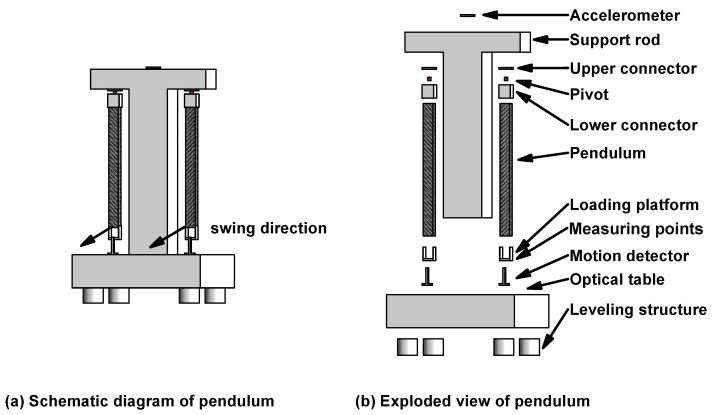
(**a**) Schematic diagram and (**b**) exploded view of MTMP.

**Figure 2 micromachines-15-00508-f002:**
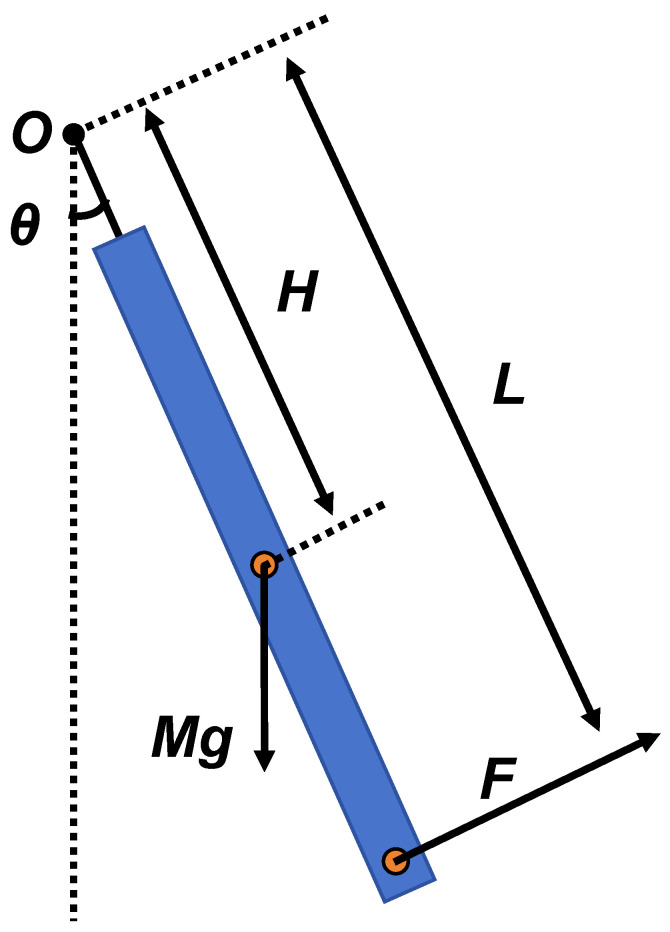
Pendulum principle schematic diagram.

**Figure 3 micromachines-15-00508-f003:**
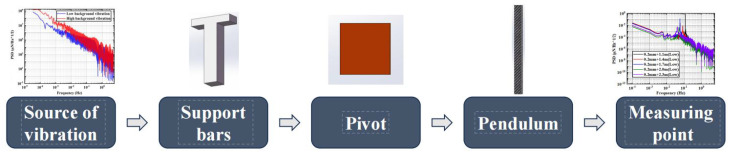
Roadmap of the vibration transmission.

**Figure 4 micromachines-15-00508-f004:**
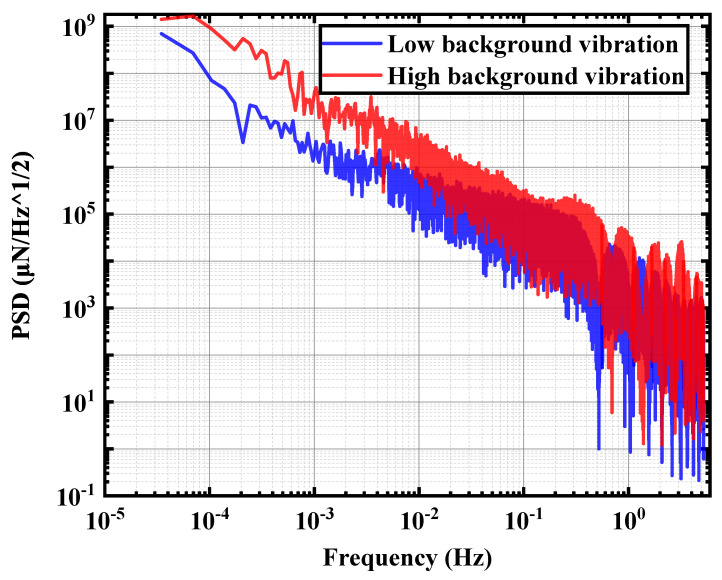
Two different ground random vibration signals.

**Figure 5 micromachines-15-00508-f005:**
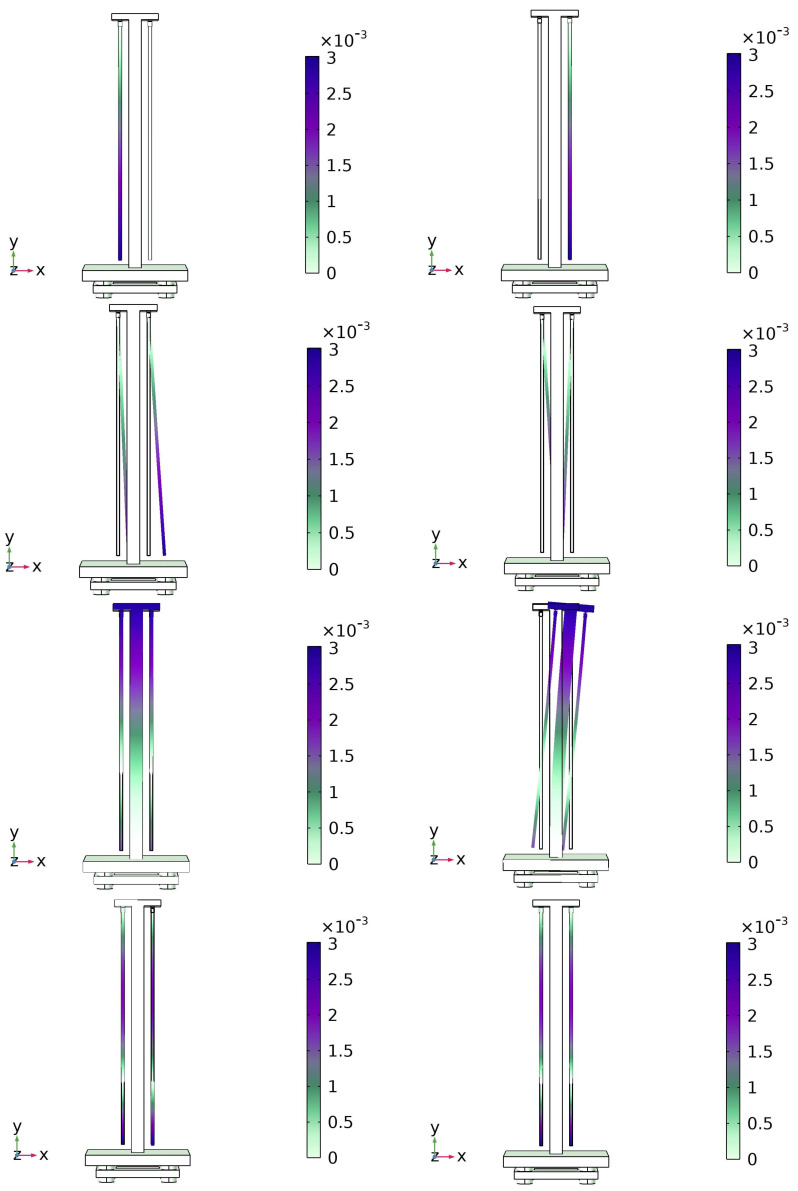
Eight modes of vibration of the model.

**Figure 6 micromachines-15-00508-f006:**
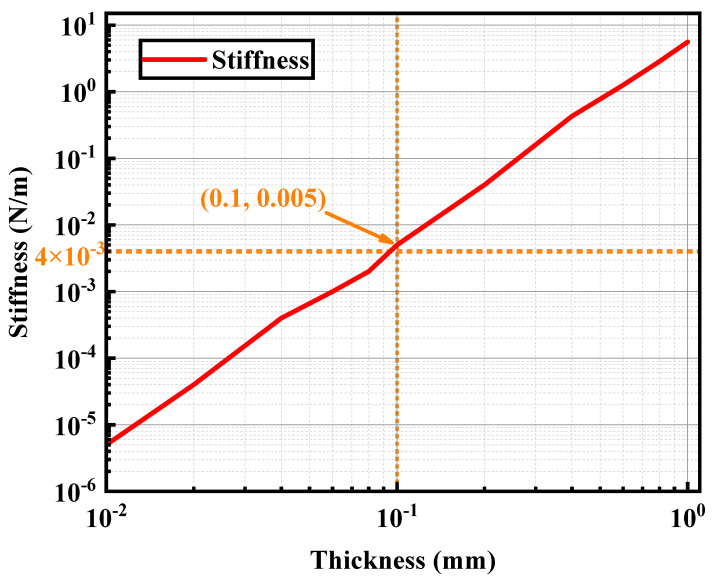
The relationship between the stiffness and thickness of sheets.

**Figure 7 micromachines-15-00508-f007:**
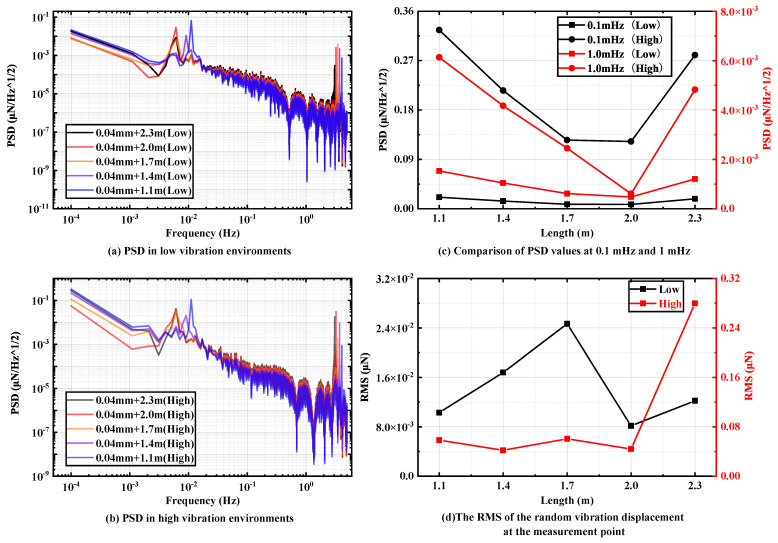
PSD of different pendulum rod lengths in (**a**) low-vibration environments and (**b**) high-vibration environments with sheet thickness of 0.04 mm. Comparison of (**c**) PSD values and (**d**) RMS values at specific frequencies.

**Figure 8 micromachines-15-00508-f008:**
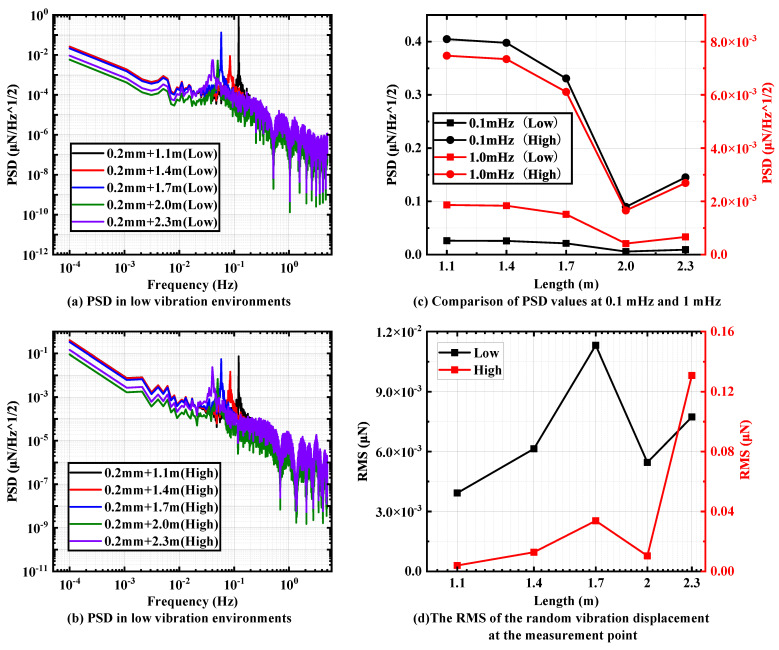
PSD of different pendulum lengths in (**a**) low-vibration environments and (**b**) high-vibration environments with sheet thickness of 0.2 mm. Comparison of (**c**) PSD values and (**d**) RMS values at specific frequencies.

**Figure 9 micromachines-15-00508-f009:**
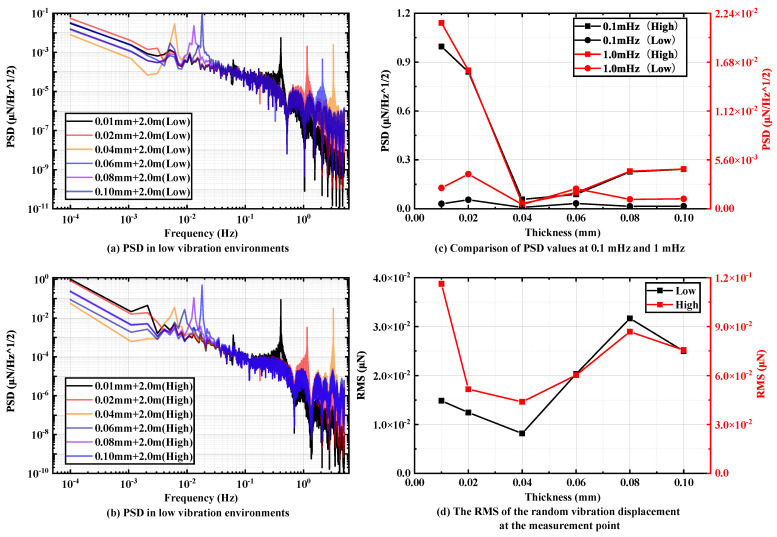
PSD of different pivot thickness (the first group) in (**a**) low-vibration environments and (**b**) high-vibration environments with pendulum rod lengths of 2 m. Comparison of (**c**) PSD values and (**d**) RMS values at specific frequencies.

**Figure 10 micromachines-15-00508-f010:**
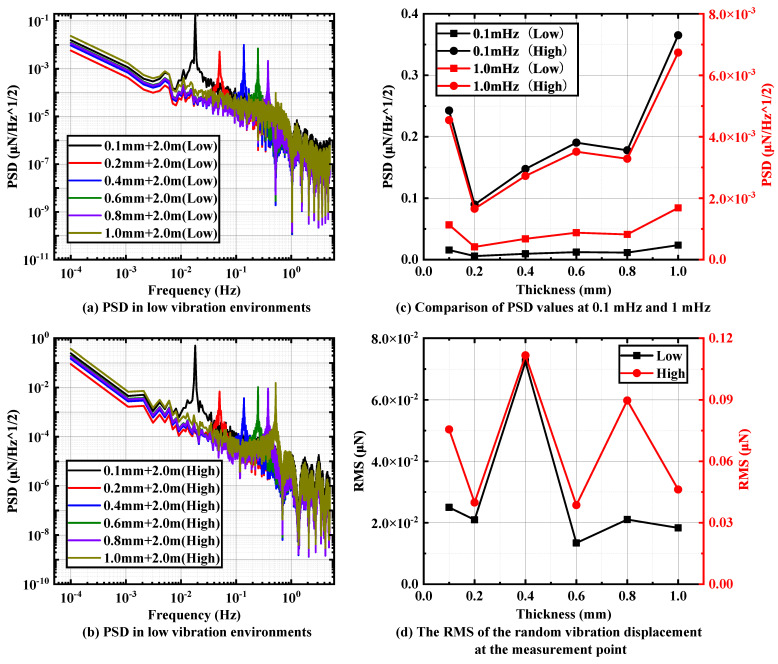
PSD of different pivot thickness (the next group) in (**a**) low-vibration environments and (**b**) high-vibration environments with pendulum rod lengths of 2 m. Comparison of (**c**) PSD values and (**d**) RMS values at specific frequencies.

**Figure 11 micromachines-15-00508-f011:**
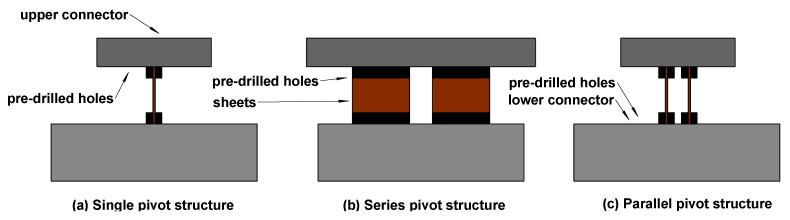
Schematic diagram of the different pivot structures: (**a**) single pivot structure, (**b**) series pivot structure, and (**c**) parallel pivot structure.

**Figure 12 micromachines-15-00508-f012:**
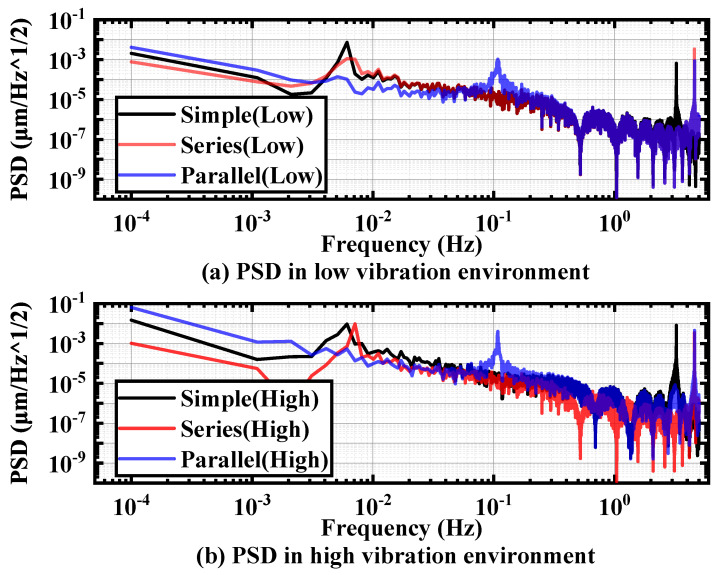
PSD values for different pivot configurations (**a**) in low-vibration environment and (**b**) in high-vibration environment.

**Figure 13 micromachines-15-00508-f013:**
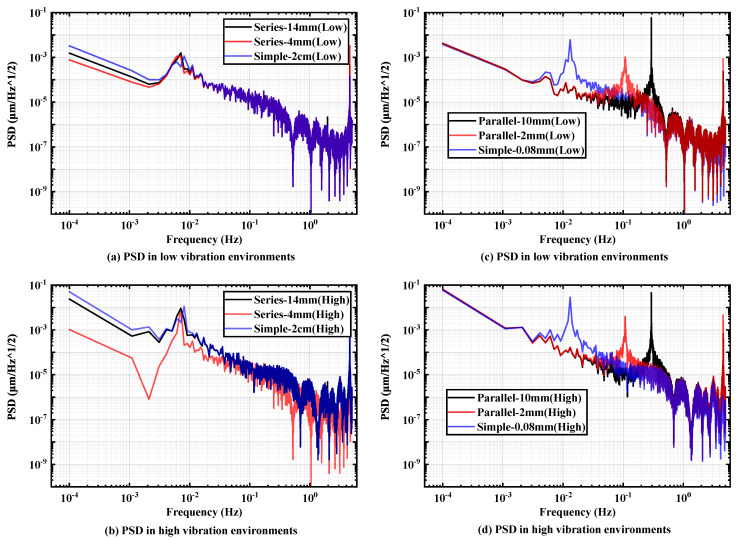
Comparison of PSD with different spacing series pivot structures (**a**) in low-vibration and (**b**) high-vibration environments. Comparison of PSD with different spacing parallel pivot structures (**c**) in low-vibration and (**d**) high-vibration environments.

**Table 1 micromachines-15-00508-t001:** Core structure parameters.

Parametric	Value
Support rod width/mm	120
Support rod length/mm	120
Support rod height/m	2.4
Pendulum rod cross section/mm^2^	30 × 30
Pendulum rod length/m	1.1, 1.4, 1.7, 2, 2.3
Thickness of sheet/mm	0.01, 0.02, 0.04, 0.06, 0.08, 0.1, 0.2, 0.4, 0.6, 0.8, 1

## Data Availability

The original contributions presented in the study are included in the article, further inquiries can be directed to the corresponding author.

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
