# Peer review of "Study of the Extremely Low-Frequency Noise Characteristics of a Micro-Thrust Measurement Platform"

_micromachines, 2024, doi:10.3390/mi15040508_

Round 1

Reviewer 1 Report

Comments and Suggestions for Authors

This study focuses on the extremely low-frequency noise characteristics of a micro-thrust measurement platform, this topic is very interesting, but I have two questions:

1. You have mentioned that “The critical structural parameters are optimized and studied using the numerical simulation method to improve the resolution and stability of the Micro-Thrust Measurement Platform (MTMP)”, But I don’t think the theoretical basis in Section 2 can support simulation and the following analysis of the extremely low-frequency noise characteristics, because the Equations (1)-(10) are all apparent in the field of vibrations. I don’t see any innovation or modification in these equations to suit the condition of extremely low-frequency noise characteristics. I do recommend that you should delete Equations (1)-(10) if you will not modify them, and that you should replace them with some quotes instead, because they are not your findings.

2.  Figure 2 should be redrawn, at least it should be clearer, or it should be in another form.

3. I do recommend you have some verification tests, unless you can confirm the boundary conditions are very complete. In my perspective, The current boundary condition in the manuscript cannot meet the requirement extremely low-frequency noise characteristics

Reviewer 2 Report

Comments and Suggestions for Authors

This manuscript focuses on developing a high-precision device for measuring weak forces. It starts by analyzing the impact of environmental vibration noise and then uses numerical simulation software to evaluate the device's stability under two different vibration scenarios. The study also examines the impact of changing the size parameter or critical structural combination of the device, intending to identify the optimal structural combination that is least affected by vibration. However, some questions should be addressed.

(1) The abstract only gives the combination of structural parameters needed to achieve the best results but does not give specific values for the achievable results, which is necessary for the abstract. The authors need to provide concise and detailed additions.

(2)The author should explain each of the physical quantities involved in the equations in the text, and there is no need to repeat the explanation once the physical quantity has appeared. In addition, it is recommended that the author should reorganize equations 6 and 7, as repeated characters may confuse the reader and misinterpret the meaning.

(3) Equation 8 is consistent with the physical phenomenon expressed in Equation 2, so it is recommended that the author delete it.

(4)On page 10, line 312, the author should check the figure being cited.

Comments on the Quality of English Language

Minor editing of English language required.

Reviewer 3 Report

Comments and Suggestions for Authors

The work deals with the development of a test bench for measuring micro-thrusts with reduced noise. The platform design is optimized by means of simulations aimed at maximizing the insulation from ground-transmitted vibrations. The following comments should be addressed.

1.      Section 2.1. The architecture of the device is unclear. Firstly, the term pendulum is used to refer to both the entire equipment and the composite rods. This may be confusing. Moreover, the device is apparently designed to work along a single direction (reasonably, horizontal and parallel to the drawing plane in Fig. 1): is it correct? Apparently, only 2 carbon fiber rods are installed, one on each side: is it correct? Which is the cross section of the rods? Where are the measuring points located? In the real application, how measurement should be performed? How the beryllium-bronze connectors are jointed to the other components? All these clarifications should be added to the section.

2.      Fig. 1. Please include a reference system in the schematic.

3.      Table 1. Reasonably, the pendulum length should be [m], not [mm].

4.      Equation 1. How is the angle theta measured? In which plane? With respect to which axis?

5.      The implementation of the COMSOL model is unclear. Apparently, the model is three-dimensional, without any restriction on the DOFs. If so, which kind of excitation is applied, i.e. along with direction? A single direction? Random excitation along all directions simultaneously? In real applications, complex modes may or may not be excited, depending on the actual operating conditions (e.g. see Refs [a-d], listed below). For instance, coupled flexural-torsional mode shapes may be excited by a single-direction excitation. These aspects should be discussed in the paper (and the mentioned works possibly added to the References, as examples), and clarifications on the actual excitation should be added.

[a]    Martini, A., et al. Structural and Elastodynamic Analysis of Rotary Transfer Machines by Finite Element Model. Journal of the Serbian Society for Computational Mechanics 2017, 11(2), 1-16. DOI: 10.24874/jsscm.2017.11.02.01

[b]    Manzato, S., et al. Wind turbine model validation by full-scale vibration test. In Proceedings of the European Wind Energy Conference (EWEC) 2010, Warsaw, Poland, April 2010

[c]    Pavlović, A., et al.  Modal analysis and stiffness optimization: The case of ceramic tile finishing. Journal of the Serbian Society for Computational Mechanics 2016, 10, 30–44.

[d]    Zaghbani, I. et al. Estimation of machine-tool dynamic parameters during machining operation through operational modal analysis. International Journal of Machine Tools and Manufacture 2009, 49(12-13), 947-957. doi: 10.1016/j.ijmachtools.2009.06.010

6.      Section 3.1. As far as the Reviewer knows, COMSOL admits acceleration signals as inputs to perform random vibration analysis. Hence, it is unclear why the seismometer signals from real environments were converted to force signals. Please, explain such choice. In any case, the analytical formulation adopted to obtain force excitation signal must be provided in this section.

7.      Which damping model has been adopted? In particular, how the damping of the carbon fiber was estimated? With reference to Section 4.1 it must be noticed that varying the length of the rods will modify non only mass and stiffness (i.e. the critical damping) but also the damping of the system, as CFRP composites can exhibit quite high damping values (e.g. see [e, f]). Moreover, the actual damping may result quite difficult to estimate, hence the results obtained without experimental validation of the numerical model being not much reliable. This appears particularly critical in the studied case, as some resonances are evidently present in the frequency range of the excitation. These aspects must be discussed in the manuscript, and the referenced papers may be added as examples.

[e]    Troncossi, M., et al. Experimental Characterization of a High-Damping Viscoelastic Material Enclosed in Carbon Fiber Reinforced Polymer Components. Appl. Sci. 2020, 10, 6193. Doi: 10.3390/app10186193

[f]      Tengfei, Q., et al. Research on damping performance of elastomer/carbon fiber epoxy composite. Mater. Res. Express 2022, 9, 020006. Doi: 10.1088/2053-1591/ac5353

8.      The mode shapes computed through the COMSOL model should be reported in the results, as they are essential for interpreting the platform response.

9.      In the optimization process, the parameters are optimized in sequence, by considering independently each parameter. This approach may not grant obtaining the solution with the global best response. Instead, starting from a sensitivity analysis taking into account all the parameters simultaneously (rod length, connector thickness, pivot structure) would be advisable. This potential issue must be discussed in the paper.

10.  Section 3.4. The pivot architecture is unclear. How are they connected to the other parts? To which part are they connected? The schematic must be improved, along with the description.

11.  Line 312. Fig. 6c is wrongly referenced.

Comments on the Quality of English Language

The length of some sentences may be reduced in order to enhance readability.

Round 2

Reviewer 1 Report

Comments and Suggestions for Authors

Thank you for revising. You have answered all my questions. 

Reviewer 3 Report

Comments and Suggestions for Authors

The review comments have been addressed quite satisfactorily.

Comments on the Quality of English Language

No further indications are provided.